# Potential Targeted Therapies in Ovarian Cancer

**DOI:** 10.3390/ph15111324

**Published:** 2022-10-26

**Authors:** Yagmur Sisman, Lau Kræsing Vestergaard, Douglas Nogueira Perez de Oliveira, Tim Svenstrup Poulsen, Tine Henrichsen Schnack, Claus Høgdall, Estrid Høgdall

**Affiliations:** 1Molecular Unit, Department of Pathology, Herlev Hospital, University of Copenhagen, DK-2730 Herlev, Denmark; 2Department of Gynecology, Juliane Marie Centre, Rigshospitalet, University of Copenhagen, DK-2100 Copenhagen, Denmark; 3Department of Gynecology, Odense University Hospital, University of Southern Denmark, DK-5000 Odense, Denmark

**Keywords:** high-grade serous carcinoma, DNA sequencing, somatic pathogenic and likely pathogenic mutations, targeted therapy, precision medicine

## Abstract

Background: We aimed to identify somatic pathogenic and likely pathogenic mutations using next-generation sequencing (NGS). The mutational findings were held against clinically well-described data to identify potential targeted therapies in Danish patients diagnosed with high-grade serous ovarian cancer (HGSC). Methods: We characterized the mutational profile of 128 HGSC patients. Clinical data were obtained from the Danish Gynecological Database and tissue samples were collected through the Danish CancerBiobank. DNA was analyzed using NGS. Results: 47 (37%) patients were platinum-sensitive, 32 (25%) partially platinum-sensitive, 35 (27%) platinum-resistant, and three (2%) platinum-refractory, while 11 (9%) patients did not receive chemotherapy. Overall, 27 (21%) had known druggable targets. Twelve (26%) platinum-sensitive patients had druggable targets for PARP inhibitors: one for tyrosine kinase inhibitors and one for immunotherapy treatment. Eight (25%) partially platinum-sensitive patients had druggable targets: seven were eligible for PARP inhibitors and one was potentially eligible for alpesilib and hormone therapy. Seven (20%) platinum-resistant patients had druggable targets: six (86%) were potentially eligible for PARP inhibitors, one for immunotherapy, and one for erdafitinib. Conclusions: PARP inhibitors are the most frequent potential targeted therapy in HGSC. However, other targeted therapies remain relevant for investigation according to our mutational findings.

## 1. Introduction

Epithelial ovarian cancer (EOC) ranks fifth in cancer deaths among women, accounting for more deaths than any other cancer of the female reproductive system (cancer.org (accessed on 15th April 2022). Denmark has the second-highest incidence rate globally, and more than 500 women are diagnosed annually [1,2]. Over 70% of cases are diagnosed in an advanced stage (International Federation of Gynecology and Obstetrics (FIGO) stage III-IV) [3]. Standard initial treatment consists of cytoreductive surgery and chemotherapy (carboplatin and paclitaxel) with the addition of bevacizumab in cases with stage IV disease or residual tumors and polyadenosine diphosphate (ADP)-ribose polymerase (PARP) inhibitors as maintenance therapy in platinum-sensitive patients. Of all patients, 20–30% are resistant to chemotherapy, and 80% showing an initial response experience relapse and ultimately die of the disease. High-grade serous ovarian cancer (HGSC) is the most common subtype. accounting for approximately 70% of EOC cases and 80% of deaths [3,4].

Increasing knowledge of the molecular, genetic, and clinical aspects underlying EOC has resulted in the development of PARP inhibitors. The PARP enzymes play a critical role in repairing DNA strand breaks. PARP inhibitors prevent the repair of these single-stranded breaks. This results in double-stranded DNA breaks, as double-strand breaks cannot be repaired accurately in tumors with homologous recombination deficiency (HRD) [5]. Ultimately, the accumulation of unrepaired DNA breaks leads to cellular death. Approximately 50% of HGSC exhibit genetic or epigenetic alterations in the homologous recombination (HR) pathway ^6^. Most often, alterations in the homologous recombination repair (HRR) pathway are caused by *BRCA* mutations. However, several other HRR genes may cause HRD, such as *ATM*, *ATR*, *CHEK1/2*, *PALB2*, and *RAD51* [6]. HRD status can be measured using different tests; as yet the Myriad MyChoice^®^ CDx test is the only FDA-approved test to determine HRD status in patients with ovarian cancer ^8^. The Myriad MyChoice^®^ CDx assay uses next-generation sequencing (NGS) to assess genomic instability, including detecting *BRCA1/2* mutations. The test measures a genomic instability score based on an algorithmic measurement composed of loss of heterozygosity, telomeric allelic imbalance, and large-scale state transitions. HRD positivity is defined as *BRCA1/2* mutations and total genomic instability score ≥42. Although the patient may not have a *BRCA1/2* alteration, they may have genomic scares resulting in HRD positivity^8^. The evidence of PARP inhibitor treatment comes from randomized clinical trials which included platinum-sensitive patients with somatic pathogenic *BRCA1/2* mutations and HRD positivity in first-line and relapsed settings [7,8]. Therefore, platinum-resistant patients do not receive PARP inhibitors in Denmark. This treatment presupposes platinum sensitivity today, which means that the treatment options are limited and a clinical challenge for these patients.

The present study aimed to identify somatic pathogenic and likely pathogenic mutations using NGS in a cohort of patients diagnosed with HGSC in order to provide an overview of possible targeted treatments based on molecular findings. The mutational findings were held against clinical data to identify potential targeted therapies in subgroups of Danish patients diagnosed with HGSC.

## 2. Results

### 2.1. Clinical Data

Clinical and pathological information for the 128 included patients is summarized in Table 1. The median age was 65 years, and 109 (85%) patients had advanced disease (FIGO III-IV). Twenty-five (19%) patients were alive at follow-up (July 2020), while 104 (81%) patients were deceased. The median overall survival (OS) was 43 months, and the median progression-free survival (PFS) was 15 months. *BRCA1/2* mutated patients had a significantly higher OS (HR: 0.50, CI 95%: 0.28–0.89) (Appendix A). Forty-seven (37%) patients were platinum-sensitive, 32 (25%) patients were partially platinum-sensitive, 35 (27%) patients were platinum-resistant, and three (2%) were platinum-refractory. Eleven patients did not receive any chemotherapy due to: dying shortly after surgery (n = 4), refraining from chemotherapy (n = 2), poor general condition (n = 2), and unknown (n = 2) (Figure 1). The excluded patients did not differ clinically from the included ones.

### 2.2. Datamining

Initially, 560,964 variants were identified. Variants not located within an exonic region were filtered out. A filter containing validity parameters such as Phred-score, p-value, allele frequency, and coverage was used. The complete list of parameters and specific criteria are outlined in Vestergaard et al. [9]. After filtering, 1349 variants were identified for further analysis. From the variants identified, we encountered the Ion Reporter^TM^ Software annotated variants from the National Center of Biotechnology Information (NCBI) ClinVar databases from 2019 (latest version 20190909). Therefore, all variants were manually inspected and cross-referenced with updated mutational verdicts from VarSome and ClinVar. We ended up with 226 likely pathogenic or pathogenic variants classified by ClinVar (Appendix A).

### 2.3. Pathogenic and Likely Pathogenic Mutations

Of the patients, 118 (92%) of 128 had one or more pathogenic or likely pathogenic mutations (Appendix A). Pathogenic and likely pathogenic mutations were distributed over 51 cancer-related genes, including *TP53* mutations in 86% of the cases (Table 2); 87%, 91%, 97%, and 67% had pathogenic or likely pathogenic mutations among platinum-sensitive, partially platinum-sensitive, platinum-resistant, and platinum-refractory patients, respectively. All the patients (n = 11) who did not receive chemotherapy had pathogenic or likely pathogenic mutations (Table 2).

### 2.4. Druggable Targets

Of the patients, 27 (21%) had potentially druggable targets distributed in ten genes, including *BRCA1/2* (16%) (Table 3 and Table 4). Thus, we found potential druggable targets in 26% of the platinum-sensitive patients (n = 12), 25% of the partially platinum-sensitive patients (n = 8), and 20% of the platinum-resistant patients (n = 7). No druggable targets were found in the platinum-refractory group (Figure 2).

### 2.5. Potential Targeted Therapies

The considered targeted therapies based exclusively on the molecular findings were PARP inhibitors (20%), tyrosine kinase inhibitors (1%), alpesilib in combination with hormone therapy (1%), erdafitinib (1%), and immunotherapy (2%) (Table 5 and Figure 2).

MSI analysis was performed in patients with *POLE* mutations or patients with five or more mutations. None of them was MSI high, and therefore they were not potential candidates for immunotherapy.

### 2.6. Mutations in the Different Platinum Response Groups

The mutational findings in the platinum response groups are listed in Appendix A and Appendix A. The platinum-sensitive, partially platinum-sensitive, and platinum-resistant patients had nine genes, including *BRCA1/2* and *TP53,* in common (Appendix A and Appendix A).

## 3. Discussion

This study characterized the mutational profile of a unique population of 128 Danish HGSC patients with corresponding tumor tissue samples and complete clinical data. The mutational profiles were characterized using Oncomine^TM^ Comprehensive Assay (OCAv3). We found potential druggable targets in 26% of the platinum-sensitive patients, 25% of the partially platinum-sensitive patients, 20% of the platinum-resistant patients, and none of the refractory cases. All platinum-sensitive patients and 88% of the partially platinum-sensitive patients with druggable targets were eligible for PARP inhibitors. Of particular interest, 86% of the platinum-resistant patients with druggable targets had targets for PARP inhibitors.

### 3.1. Mutation Profile in Danish Ovarian Cancer Patients Compared to Other Studies

The most extensive study based on data from the TCGA database showed *TP53* mutations in 88% and *BRCA1/2* mutations in 23% of 316 HGSC tumors [10]. Patch et al. characterized the mutational profile of 80 HGSC patients [11]. They found *TP53* mutations in 99% and *BRCA1/2* mutations in 26%. The mutational differences regarding *TP53* and *BRCA1/2* in our study compared with TCGA and Patch et al. may have several explanations. First, our research had a therapeutic focus, opting to include only pathogenic or likely pathogenic mutations according to the American College of Medical Genetics and Genomics (ACMG) classification of variants [12]. Ten (8%) patients did not have pathogenic or likely pathogenic mutations. However, except for one, all these patients had Variants of Unknown Significance (VUS), which are dynamic variants that may change classification in the future. Neither TGCA nor Patch et al. has used the ACMG variant classification, as these studies had a broader focus on characterizing the overall mutational profile. Therefore, it is not surprising that they report a higher frequency of *TP53* and *BRCA1/2* mutations.

Differences in patient selection may be another explanation. In the study by Patch et al., 39% were platinum-sensitive and 61% were platinum-resistant or -refractory, while TCGA included 69% platinum-sensitive and 31% platinum-resistant patients [10,11]. In TCGA and Patch et al., platinum status was defined as resistant if the patient had progressed or recurred within 6 months after the end of the last treatment, which is consistent with our definition. Moreover, we defined relapse and PD from the best clinical evaluation based on CT/MRI/PET-CT scans, serum CA125, and patients’ symptoms. The follow-up in the two published studies is not fully described.

Finally, we used targeted sequencing of all cancer-related genes, while TCGA used whole-exome sequencing (WES) and Patch et al. used whole-genome sequencing (WGS). WES identifies variants within all coding regions, and WGS identifies variants in the whole genome. Both WES and WGS generate massive data without therapeutic relevance. In this study, we had a clinical focus on targeted therapies. Therefore, we used OCAv3, which sequences all known druggable targets today. Panel sequencing is the most cost-effective method. It can easily be implemented in a routine diagnostic setting with a fast throughput, which meets the demand for a cancer fast-track package. Furthermore, it has a higher depth, increases the statistical power of detecting variants, and lowers variant detection limits. Nonetheless, panels are dynamic and change regularly with increased knowledge about diseases to remain clinically relevant [13,14].

In 2008 Soegaard et al. established the prevalence of *BRCA1*/*2* mutations in a population-based study of 445 EOC cases from Denmark [15]. They identified pathogenic mutations in 26 patients (5.8%), 22 mutations in *BRCA1* (4.9%) and four in *BRCA2* (0.9%)), compared with 16.4% in our study. In contrast to our study, only 62% of the included patients were diagnosed with serous adenocarcinomas. Furthermore, only tumors in the BRCA mutated patients were graded [15]. The study by Soegaard et al. used Sanger sequencing, while we used NGS. The sensitivity for these two methods is 20% versus 5% (Sanger/NGS), which may explain the lower frequency of reported *BRCA1/2* mutations in the study by Soegaard et al. [13]. Furthermore, the identified *BRCA1*/*2* mutations differed in the two studies, possibly because more pathogenic *BRCA1/2* mutations have been identified since 2008.

### 3.2. Platinum Resistance and PARP Inhibitors

Based on the molecular findings, we found that six platinum-resistant patients (17%) may potentially be eligible for PARP inhibitors. Three patients had a somatic pathogenic or likely pathogenic *BRCA1/2* mutation, and three patients had mutations in the *ATM*, *CHEK1*, or *CDK12* genes. However, due to the European and American approvals, none of them would be offered treatment with PARP inhibitors as standard treatment [16,17]. In Denmark, PARP inhibitors are only approved as maintenance treatment for newly diagnosed advanced EOC with a *BRCA1/2* mutation and partial or complete response to platinum-based chemotherapy. In the recurrence setting, only platinum-sensitive *BRCA* mutated or HRD patients are candidates for PARP inhibitor treatment [18].

To date, limited data is available on the activity of PARP inhibitors in platinum-resistant HGSC. No studies have examined the effect in platinum-resistant patients with mutations in genes other than *BRCA1/2* (Appendix A) [19,20,21,22,23,24]. The BAROCCO trial investigated the combination of olaparib and cediranib in 123 HGSC platinum-resistant patients [23]. They found a trend toward improved PFS (5.7 months vs. 3.1 months). In the ARIEL4 trial (NCT02855944), rucaparib was tested in 345 *BRCA* mutated patients, of whom 176 were platinum resistant. PFS was significantly higher in the rucaparib group [24]. However, the PFS has not yet been calculated specifically for platinum-resistant patients. Results are pending from the NRG-GY005 (NCT02502266) and OCTAVIA (NCT03117933) trials, which are ongoing studies comparing PARP inhibitors (alone or in combination with cedirabin) with platinum-free chemotherapy in platinum-resistant patients (Appendix A). Few studies have examined whether platinum-resistant patients benefit from PARP inhibitors. Nevertheless, 17% of our platinum-resistant patients might be potentially eligible for PARP inhibitors based on their mutational profiles. PARP inhibitors represent a promising trend in response rates and PFS in platinum-resistant patients. However, results from more extensive high-quality randomized studies are needed. Unfortunately, the present cohort was not treated with PARP inhibitors; thus, response and survival estimates cannot be assessed.

Our study has several strengths. All patients were operated on and treated at the same tertiary center by gynecologic oncologists with high expertise. All clinical data were continuously updated in the Danish Gynecologic Database, and none of the patients were lost to follow-up. Furthermore, our cohort size is comparable with TCGA and Patch et al. [10,11]. However, there are limitations as well. First, PARP inhibitors may be effective in more patients than those identified here, as HRD assessment might find additional patients who would benefit from PARP inhibitors. Moreover, although the cohort is relatively large, the subgroups that can be offered potential targeted therapies based solely on molecular findings are small, resulting in statistical limitations (Figure 2). Lastly, the mutational findings in the platinum response subgroups confirm that HGSC is a highly heterogenous molecular disease (Appendix A). The number of patients with specific mutations in the subgroups was too small to perform statistical analyses (Appendix A). Therefore, it was not possible to define molecular profiles in the platinum response subgroups based on our findings.

PARP inhibitors are the most frequent potential targeted therapy for HGSC patients, according to our molecular findings. This confirms the results of previous studies and the clinical practice today [10,25]. However, it is highly relevant to investigate other targeted therapies, especially in platinum-resistant and platinum-sensitive patients who no longer respond to PARP inhibitors. Therefore, extensive studies which can clarify the efficacy of PARP inhibitors in platinum-resistant patients are needed to identify molecular druggable targets. Furthermore, large basket trials such as NCI-MATCH and ProTarget with the inclusion of HGSC patients with druggable targets other than those known for PARP inhibitors are warranted. Studies investigating the efficacy of combination therapies must be prioritized to overcome resistance. Lastly, most HGSC patients have a *TP53* mutation. Efficient treatments targeting *TP53* can potentially improve the treatment of ovarian cancer tremendously. APR-246 is a drug that reactivates mutant p53 encoded by *TP53* in cancer cells by promoting its correct wild-type folding [26,27]. However, the effect of APR-246 has not yet been examined in a large clinical trial. A Phase 1/2 Open-label, Multicenter Study is ongoing to Assess the Safety, Tolerability, Pharmacokinetics, Pharmacodynamics, and Efficacy of PC14586 in Patients with Advanced Solid Tumors Harboring a p53 Y220C Mutation (PYNNACLE). Although only a few patients have this specific mutation, it points toward new perspectives and directions in the future treatment of HGSC.

## 4. Materials and Methods

### 4.1. Patients and Tissue Samples

One hundred and twenty-eight out of 157 patients with a primary diagnosis of HGSC were selected from a retrospective cohort (Figure 1) [28,29]. Sequencing was conducted with DNA from the patients. Patients were registered in the Pelvic Mass study/GOVEC study, a cohort initiated in September 2004. The clinical information of the patients was registered in the Danish Gynecological Cancer Database (DGCD) [30]. Platinum resistance was defined as relapse or progressive disease (PD) within six months after chemotherapy. Patients who developed PD during treatment or within four weeks after the last cycle were considered platinum-refractory. Patients were considered platinum-sensitive if they had experienced no relapse or PD or if relapse occurred more than six months after the end of first-line chemotherapy. Relapse and PD were defined from the best clinical evaluation based on CT/MRI/PET-CT scans, serum CA125, and patients’ symptoms. Patients were followed from October 2004 until July 2020 with a minimum follow-up of 10 years. No patients were lost to follow-up. All tissue samples were registered and stored in the Danish Cancer Biobank (DCB, Bio- and GenomeBank, Denmark) and handled according to national guidelines (www.regioner.dk/rbgb (accessed on 1 September 2020). The cohort was examined and classified as HGSC by a pathologist specialized in gynecologic oncology.

### 4.2. DNA Extraction

A representative area of cancerous tissue in FFPE blocks was identified and extracted to ensure the high content of tumor cells by an experienced pathologist within oncogynecology. Genomic DNA was extracted using Maxwell^®^ RSC DNA FFPE Kit (Promega). DNA concentration was quantified using the Qubit^TM^ ds DNA High-Sensitive Assay kit (Thermo Fisher Scientific, Waltham, MA, US) on the Qubit fluorometer (Thermo Fisher Scientific, Waltham, MA, US).

### 4.3. Library Preparation and Sequencing

Library preparation was performed manually for the OCAv3. The assay consisted of 161 cancer-related genes with the sequencing of coding areas of all known druggable targets (Thermo Fisher Scientific) [12]. PCR amplification was conducted using a DNA concentration of 20 ng as input. Sequencing was performed using the Ion S5^TM^ XL Sequencer (Thermo Fisher Scientific). The Ion Reporter^TM^ Software (v. 5.14) was used for initial automated analysis, and Oncomine Comprehensive v3–w4.0–DNA–Single Sample was used for analysis workflow.

### 4.4. Microsatellite Instability (MSI)

According to the manufacturer’s instructions, microsatellite instability (MSI) testing was performed using the PlentiPlexTM MSI PentaBase panel (v.1.9.2); 20 ng/μL extracted DNA per patient was used for analyses. MSI was evaluated individually by comparing the length of amplicons obtained in the patients’ tumor-derived DNA with the reference DNA by molecular biologists.

### 4.5. Data Analysis

All data analyses were conducted using Python programming language (v.3.7). Data were initially pre-filtered considering variants within exonic regions or splice site regions and classified as either single-nucleotide variants, multi-nucleotide variants, or indels. Variants passing the pre-analysis data cleaning were subsequently analyzed and annotated based on the variant filtering properties outlined in Vestergaard et al. [9]. Variants were clinically annotated using the database provided at VarSome “https://varsome.com/ (access on 13–14 October 2021). using the implemented ClinVar database. Only variants classified as *pathogenic* and *likely pathogenic* by expert panels on ClinVar were included. Files for analysis were provided from the Ion Reporter^TM^ Software (v. 5.14). Files were downloaded without any filter chain, providing all identified variants. Potential druggable targets were known oncogenes. The observed genetic alterations were class 4 or 5 (according to ACMG variant classification [12]) with the well-known effect of an FDA-approved targeted therapy. The Ion Torrent^TM^ Oncomine^TM^ Reporter (Thermo Fisher Scientific v.5.3.0) was used for the search. The reports from Ion Torrent^TM^ Reporter with the clinical evidence grading are included in Appendix A. All *pathogenic* and *likely pathogenic* variants were reviewed on 25th November 2021 with the Demo US and EU and Global Clinical Trials filters.

As immunotherapy recommendations are not reported in the Ion Torrent^TM^ Oncomine^TM^ Reporter, these recommendations were found in the ESMO guidelines (25NOV2021) [25].

### 4.6. Statistical Analysis

All statistical analyses were performed with Python programming language (v. 3.7). Fisher’s Exact Test was used to calculate and determine potential strand bias.

## 5. Conclusions

PARP inhibitors are the most frequent potential targeted therapy in HGSC. However, according to our mutational findings, most patients were not candidates for PARP inhibitors. Interestingly, we found several pathogenic mutations in most of the examined tumors. Although these mutations cannot be targeted today, they may become so in the future with the development of new treatment strategies. This is important to all those not eligible for PARP inhibitors as well as in the recurrent setting, which is often characterized by multi-chemotherapy resistance. Therefore, extensive studies investigating the efficacy of PARP inhibitors in platinum-resistant patients along with basket trials with the inclusion of HGSC patients with druggable targets other than those known for PARP inhibitors are needed.

## Figures and Tables

**Figure 1 pharmaceuticals-15-01324-f001:**
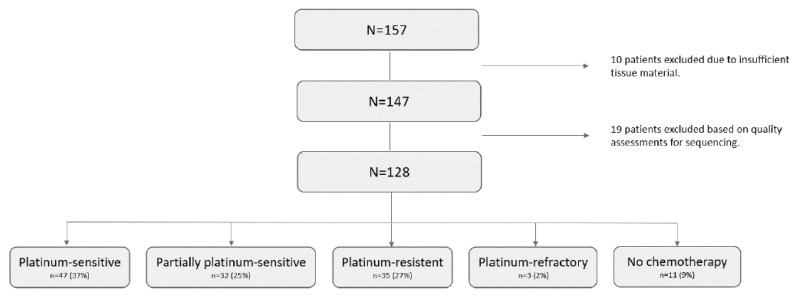
Flow diagram of patient inclusion.

**Figure 2 pharmaceuticals-15-01324-f002:**
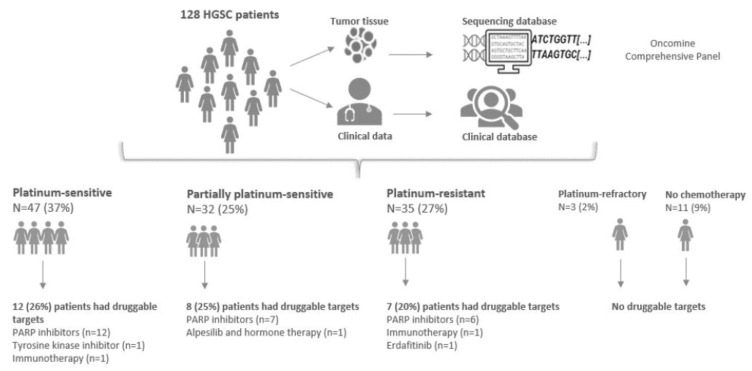
Potential targeted therapies in HGSC.

**Table 1 pharmaceuticals-15-01324-t001:** Baseline characteristics of 128 patients diagnosed with HGSC.

Median age in year	65 (41–89)
Median CA125	792 (9–17.160)
Median RMI	4632 (63–153.432)
Median BMI	24 (15–48)
Median follow up time in months	92 (61–123)
**Performance score**	
0	54 (42%)
1	52 (41%)
2	18 (14%)
3	4 (3%)
**FIGO stage**	
I	9 (7%)
II	10 (8%)
III	89 (70%)
IV	20 (15%)
**Residual tumor after surgery**	
0	50 (39%)
<1 cm	24 (19%)
>1 cm ≤ 2 cm	17 (13%)
>2 cm	37 (29%)
**Chemotherapy (n = 117)**	
Carboplatin and taxol	113 (97%)
Carboplatin	4 (3%)
**Chemotherapy response**	
>12 months (sensitive)	47 (40%)
6 - ≤ 12 months (partially platinum-sensitive)	32 (27%)
≤6 months (resistant)	35 (30%)
Refractory	3 (3%)
**OS in months**	
OS (median)	43 (0.7–190)
OS FIGO stage I (median)	138 (45–184)
OS FIGO stage II (median)	54 (3–140)
OS FIGO stage III (median)	38 (0.7–190)
OS FIGO stage IV (median)	33 (0.8–180)
**PFS in months**	
PFS (median)	15 (0–118)
PFS FIGO stage I (median)	74 (23–118)
PFS FIGO stage II (median)	19 (0–74)
PFS FIGO stage III (median)	15 (0–107)
PFS FIGO stage IV (median)	10 (0–95)

**Table 2 pharmaceuticals-15-01324-t002:** Genes with pathogenic or likely pathogenic mutations in 128 HGSC patients.

**All patients (n = 128)**	*ARID1A* *ATM* * *BAP1* *BRCA1* ** *BRCA2* ** *CDK12* * *CHEK1* * *CREBBP* *CTNNB1* *DDR2* *EGFR* * *FANCA* *FANCI*	*FBXW7* *FGFR3* * *FGFR4* *FLT3* *FOXL2* *H3F3A* *KRAS* *MAP2K1* *MAX* *MLH1* * *MRE11* *MSH2* * *MTOR*	*MYC* *NF1* *NF2* *NOTCH1* *NOTCH2* *NOTCH3* *NTRK2* *NTRK3* *PDGFRB* *PIK3CA* * *PIK3CB* *POLE* *PTCH1*	*PTPN11* *RAD51* *RB1* *SETD2* *SLX4* *SMARCA4* *SMARCB1* *SMO* *TERT* *TP53* (86%) *TSC1* *TSC2*
**Platinum-sensitive** **(n = 47)**	*ARID1A* *ATM* * *BRCA1* ** *BRCA2* ** *CDK12* * *CREBBP* *EGFR* *	*FANCA* *FGFR4* *FLT3* *H3F3A* *KRAS* *MAP2K1* *MSH2* *	*NF1* *NOTCH1* *NOTCH3* *PDGFRB* *PIK3CB* *PTCH1* *RAD51*	*RB1* *SMARCA4* *SMARCB1* *TP53* (85%) *TSC1* *TSC2*
**Partially platinum-sensitive (n = 32)**	*ARID1A* *BRCA1* ** *BRCA2 *** *CREBBP*	*FANCA* KRAS *NF1*	*NOTCH1* *NOTCH3* *PIK3CA **	*POLE* *SMO* *TP53* (91%)
**Platinum-resistant (n = 35)**	*ATM* * *BAP1* *BRCA1* ** *BRCA2* ** *CDK12* * *CHEK1* * *CREBBP* *CTNNB1* *DDR2* *FANCA*	*FANCI* *FBXW7* *FGFR3* * *FOXL2* *KRAS* *MAX* *MLH1* * *MRE11* *MTOR*	*MYC* *NF1* *NF2* *NOTCH1* *NOTCH2* *NTRK2* *NTRK3* *PDGFRB* *POLE*	*PTPN11* *RAD51* *RB1* *SETD2* *SMARCA4* *SMARCB1* *TERT* *TP53* (83%) *TSC1*
**Platinum-refractory (n = 3)**	*ARID1A*	*CREBBP*	*TP53* (33%)	
**No chemotherapy** **(n = 11)**	*NF1*	*NOTCH1*	*SLX4*	*TP53* (100%)

* Druggable target in other cancers. ** Druggable target in ovarian cancer.

**Table 3 pharmaceuticals-15-01324-t003:** Genes with druggable targets (Nov 2021).

Gene	Cancer	Therapy
*ATM*	Prostata cancer	Olaparib
*BRCA1*	Prostate cancer	Bevacizumab + olaparib, niraparib, olaparib, rucaparib
*BRCA2*	Prostate cancer	Bevacizumab + olaparib, niraparib, olaparib, rucaparib
*CDK12*	Prostate cancer	Olaparib
*CHEK1*	Prostate cancer	Olaparib
*EGFR*	Non-small cell lung cancer	Afatinib, bevacizumab + erlotinib, bevacizumab + gefitinib, dacominitib, erlotinib + ramucirumab, gefitinib, gefitinib + chemotherapy.
*FGFR3*	Bladder cancer, bladder urothelial carcinoma	Erdafitinib
*MLH1* *MSH2*	Anaplastic thyroid cancer, bladder cancer, brain metastases from solid tumors, colorectal cancer, cutaneous melanoma, hepatocellular carcinoma, hereditary gastrointestinal cancers, hypopharyngeal cancer, larynx cancer, lung and malignant pleural mesothelioma, non-small cell lung cancer, oral cavity cancer, renal cell carcinoma, small-cell lung cancer	Immunotherapy
*PIK3CA*	Breast cancer	Alpelisib and hormone therapy

**Table 4 pharmaceuticals-15-01324-t004:** Druggable targets in HGSC patients.

	Platinum-sensitive (N = 47)	Partially platinum-sensitive (N = 32)	Platinum-resistant (N = 35)
*BRCA1/2*	n = 10 *BRCA1* p.Leu1216PhefsTer2 *BRCA1* p.Asp1305AlafsTer2 *BRCA1* p.Arg1699Gln *BRCA1* p.Gln494Ter *BRCA1* p.Glu1134Ter *BRCA1* p.Gln1756PorfsTer74 *BRCA1* p.? splice site mutation *BRCA1* p.Arg1699Gln *BRCA1* p.Gln1756ProfsTer74 *BRCA2* p.Trp2626Cys	n = 7 *BRCA1* p.? splice site mutation *BRCA1* p.Ala1708Glu *BRCA1* p.Gly1738Glu *BRCA1* p.Asn609IlefsTer3 *BRCA2* p.Ser1741ThrfsTer35 *BRCA2* p.Ser2219Ter *BRCA2* p.Lys2909GlnfsTer16	n = 2 *BRCA1* p.Glu1046Ter *BRCA1* p.Glu23ValfsTer17
*BRCA1*/2 and other druggable targets	n = 1 *BRCA1* p.? splice site mutation *ATM* p.Arg2598Ter *EGFR* p.Ala289Thr *MSH2* p.Gln324Ter *MSH2* Trp345Ter *MSH2* p.Gln413Ter		n = 1 *BRCA2* p.Trp3106Ter *FGFR3* p.Trp685Ter
Other druggable targets	n = 1 *CDK12* p.Ser301CysfsTer5	n = 1 *PIK3CA* p.Glu542Lys	n = 4 *ATM* p.Glu1751Ter *CDK12* p.Gln602Ter *CHEK1* p.Gln318Ter *MLH1* p.Arg659Ter

**Table 5 pharmaceuticals-15-01324-t005:** Potential targeted therapies in HGSC patients.

	Platinum-sensitive (N = 47)	Partially platinum-sensitive (N = 32)	Platinum-resistant (N = 35)
PARP inhibitor	n = 12 *BRCA1* chr17:41243899 p.Leu1216PhefsTer2 *BRCA1* chr17:41243633 p.Asp1305AlafsTer2 *BRCA1* chr17:41215947 p.Arg1699Gln *BRCA1* chr17:41246068 p.Gln494Ter *BRCA1* chr17:41244148 p.Glu1134Ter *BRCA1* chr17:41209079 p.Gln175ProfsTer74 *BRCA1* chr17:41215969 p.? *BRCA1* chr17:41215947 p.Arg1699Gln *BRCA1* chr17:41209079 p.Gln1756ProfsTer74 *BRCA1* chr17:41209153 p.? *BRCA2* chr13:32936732 p.Trp2626Cys *ATM* chr11:108203492 p.Arg2598Ter *CDK12* chr17:37619224 p.Ser301CysfsTer5	n = 7 *BRCA1* chr17:41222944 p.? *BRCA1* chr17:41215920 p.Ala1708Glu *BRCA1* chr17:41209133 p.Gly1738Glu *BRCA1* chr17:41245721 p.Asn609IlefsTer3 *BRCA2* chr13:32950896 p.Lys2909GlnfsTer16 *BRCA2* chr13:32915148 p.Ser2219Ter *BRCA2* chr13:32913710 p.Ser1741ThrsfTer35	n = 6 *BRCA1* chr17:41244412 p.Glu1046Ter *BRCA1* chr17:41276044 p.Glu23ValfsTer17 *BRCA2* chr13:32968887 p.Trp3106Ter *ATM* chr11:108172448 p.Glu1751Ter *CDK12* chr17:37627889 p.Gln602Ter *CHEK1* chr11:125514014 p.Gln318Ter
Tyrosine kinase inhibitor	n = 1 *EGFR* chr7:55221821 p.Ala289Thr		
Immunotherapy	n = 1 *MSH2* chr2:47643462 p.Gln324Ter *MSH2* chr2:47643526 p.Trp345Ter *MSH2* chr2:47657041 p.Gln413Ter		n = 1 *MLH1* chr3:37090086 p.Arg659Ter
Alpesilib and hormone therapy		n = 1 *PIK3CA* chr3:178936082 p.Glu542Lys	
Erdafitinib			n = 1 *FGFR3* chr4:1808296 p.Trp685Ter

## Data Availability

Data is contained within the article.

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
