# Peer review of "Potential Targeted Therapies in Ovarian Cancer"

_pharmaceuticals, 2022, doi:10.3390/ph15111324_

Round 1

Reviewer 1 Report

The author has presented significant research findings aimed to identify the somatic pathogenic and likely pathogenic mu- 61 mutations using next-generation sequencing (NGS). However, the author needs to rewrite the result section with more clarity on his/her research findings. Also, the author needs to replace the word conclusions with the conclusion in the abstract section.

Explain the significance of the results obtained in section 2.4.

The future aspect of the study is also not very clearly presented. Hence the author is advised to provide a strong conclusion of the study in the abstract section.

Research involves patient data, hence I would like to ask the author to provide an ethical clearance of the research study.

Reviewer 2 Report

Introduction:

Importance and definition of HRD and HRD genes/testing is not enough clarified in the introduction. 

Results & discussion: 

strength of the study: availability of clinical data from HGSC patients with corresponding tumor tissue samples 

However, no statistics were used to compare the different groups. 

A summarizing figure to present the most significant results would be an added value.

How do the authors define a "potential drugable target" (cfr suggestions in the methods section)?

methods:

-There is a lack of explaining whether a variant is classified as drugable or not, also levels of clinical evidence are lacking. (FDA, EMA, ESMO, clinical trials, pre-clinical trials

- statistical analaysis is done in Python, however no statistical tests are explained for example differences between groups ( platinum sensitive-refractory-resistant)

Round 2

Reviewer 2 Report

The authors clearly made an effort to adapt the manuscript according to reviewers comments within their capacity of obtained results. 

I was not able to find figure 2, therefore I can not judge. It should give a clear overview of the patient cohort and the assigned mutational profiles with a clear visualization of which mutations are druggable and which are not (or only potentially druggable as described in other cancer types) .

Since the adaptation of the manuscript a lot of text editing will be necessary due to typo's.

Author Response

Dear reviewer, 

Thank you for your feedback and help. The quality of our work improved with your help. 

Figure 2 is uploadet in first revision. We don't now why you can't see it but we have contacted the journal to make sure that it will be a part of the manuscript. Figure 2 is an overview of the patient group, the platinum groups with the druggable targets in each group and thereby the potential targeted therapies.

Text editing have now taken place with the help of the writing program Grammarly Pro. 

Hope that you like the new version of the manuscript. Thanks again for your help.